# Highly Sensitive and Selective Formaldehyde Gas Sensors Based on Polyvinylpyrrolidone/Nitrogen-Doped Double-Walled Carbon Nanotubes

**DOI:** 10.3390/s22239329

**Published:** 2022-11-30

**Authors:** Thanattha Chobsilp, Thotsaphon Threrujirapapong, Visittapong Yordsri, Alongkot Treetong, Saowaluk Inpaeng, Karaked Tedsree, Paola Ayala, Thomas Pichler, Lei Shi, Worawut Muangrat

**Affiliations:** 1Department of Materials and Production Technology Engineering, Faculty of Engineering, King Mongkut’s University of Technology North Bangkok, Bangkok 10800, Thailand; 2National Metal and Materials Technology Center, Pathumthani 12120, Thailand; 3National Nanotechnology Center, Pathumthani 12120, Thailand; 4Department of Chemistry, Faculty of Science, Burapha University, Chonburi 20131, Thailand; 5Electronic Properties of Materials, Faculty of Physics, University of Vienna, 1090 Vienna, Austria; 6School of Materials Science and Engineering, Sun Yat-sen University, Guangzhou 510275, China; 7Department of Advanced Materials Engineering, Faculty of Engineering, Burapha University, Chonburi 20131, Thailand

**Keywords:** nitrogen-doped double-walled carbon nanotubes, polyvinylpyrrolidone, formaldehyde, gas sensors

## Abstract

A highly sensitive and selective formaldehyde sensor was successfully fabricated using hybrid materials of nitrogen-doped double-walled carbon nanotubes (N-DWCNTs) and polyvinylpyrrolidone (PVP). Double-walled carbon nanotubes (DWCNTs) and N-DWCNTs were produced by high-vacuum chemical vapor deposition using ethanol and benzylamine, respectively. Purified DWCNTs and N-DWCNTs were dropped separately onto the sensing substrate. PVP was then dropped onto pre-dropped DWCNT and N-DWCNTs (hereafter referred to as PVP/DWCNTs and PVP/N-DWCNTs, respectively). As-fabricated sensors were used to find 1,2-dichloroethane, dichloromethane, formaldehyde and toluene vapors in parts per million (ppm) at room temperature for detection measurement. The sensor response of N-DWCNTs, PVP/DWCNTs and PVP/N-DWCNTs sensors show a high response to formaldehyde but a low response to 1,2-dichloroethane, dichloromethane and toluene. Remarkably, PVP/N-DWCNTs sensors respond sensitively and selectively towards formaldehyde vapor, which is 15 times higher than when using DWCNTs sensors. This improvement could be attributed to the synergistic effect of the polymer swelling and nitrogen-sites in the N-DWCNTs. The limit of detection (LOD) of PVP/N-DWCNTs was 15 ppm, which is 34-fold higher than when using DWCNTs with a LOD of 506 ppm. This study demonstrated the high sensitivity and selectivity for formaldehyde-sensing applications of high-performance PVP/N-DWCNTs hybrid materials.

## 1. Introduction

Formaldehyde (HCHO), one of the most dangerous volatile organic compounds (VOCs), is a colorless, highly toxic and flammable chemical. Formaldehyde has been used in building materials and household products (e.g., glues, lacquers, paints, and coatings). In addition, it is emitted from fuel burning appliances and cigarette smoke. It can cause many symptoms, such as irritation of the eyes, nose, throat and skin, as well as coughing, nausea. Coughing and nausea can be caused by short-term exposure and can be carcinogenic, and there is a risk of leukemia from long-term exposure [1,2,3]. Thus, the accurate monitoring of formaldehyde is a serious consideration. Various conventional techniques are available for the detection of formaldehyde such as high-performance liquid chromatography (HPLC) and gas chromatography-mass spectrometry (GC-MS). Both HPLC and GC-MS are powerful analytical tools for the analysis of complex gas phase mixers which have high sensitivity, selectivity, and reliability [4,5,6]. However, the disadvantage of these techniques is that they are high-cost, complex, time-consuming and impracticable for real-time monitoring, and need to be carried out by skilled, expert technicians.

Chemical sensors based on carbon nanotubes (CNTs) are developed and applied as sensing materials because of their unique structure and excellent properties [7,8,9,10,11]. However, pristine CNTs are still insensitive to formaldehyde. Efforts have been made to develop the surface modification and nanocomposites of CNTs with metal oxides for improving formaldehyde detection [12,13,14,15]. For example, Wang et al. [13] fabricated tin oxide-doped with hydroxyl functionalized MWCNTs (MWCNTs-doped SnO_2_) for formaldehyde detection at approximately 125 to 375 °C. The results showed that the most effective operating temperature was 250 °C. The lowest concentration of formaldehyde detected by MWCNTs-doped SnO_2_ at 250 °C was 0.03 parts per million (ppm). Although the response of the MWCNTs-doped SnO_2_ sensor was about three times higher than that of an undoped SnO_2_ sensor at 250 °C, the drawback of metal oxide-based gas sensors is that they require operation at high temperature to achieve high performances. Xie et al. [14] prepared sensors with amino-functionalized multi-walled CNTs (amino group-MWCNTs) for formaldehyde detection at room temperature. Pure MWCNTs are hardly sensitive to formaldehyde. The relative resistance change of amino group-MWCNTs is about 3–13 times that of pure MWCNTs. The sensor represented high sensitivity of formaldehyde because of the number of amino groups modified on the MWCNTs. The limit of detection (LOD) of amino group-MWCNTs was 20 parts per billion (ppb) with a response value of 1.79%. Amino group-MWCNTs sensors exhibited good performance for formaldehyde, but the preparation process of the functionalization of commercial MWCNTs with amino groups involved numerous functionalization methods (e.g., refluxing, evaporating and centrifugation several times), which is not economically attractive. 

Polymer is one of the sensing materials for VOC detection due to its high sensitivity, short response and the fact that it operates at room temperature. Polyvinylpyrrolidone (PVP) is a low-cost polymer with highly interesting physical and chemical properties, such as high thermal and chemical stability, non-toxicity and excellent solubility in many organic solvents [16], which make PVP suitable materials for VOCs detection. Gas sensors based on CNTs-PVP composite films for VOCs detection at room temperature were proposed by Xie et al. [17]. CNTs-PVP composite sensors show a high sensitivity and selectivity to 1,2-dichloroethane vapor. Nevertheless, there have been very few reports of formaldehyde detection based on CNTs-PVP hybrid materials. Recently, the surface modification of CNTs using heteroatoms, such as boron and nitrogen atoms, in the CNTs lattice were proposed in order to adjust their electronic properties. Based on experimental results, nitrogen-doped CNTs improve the response to ammonia, nitrogen dioxide and carbon monoxide gases, while boron-doped CNTs improve the response to ethylene gas [18,19,20]. However, to date there are no reports on experimental studies of nitrogen-doped CNTs for formaldehyde detection. 

In this study, we proposed using polymer-CNTs hybrid materials for formaldehyde detection at room temperature. Nitrogen-doped double-walled CNTs (N-DWCNTs) were produced by high-vacuum chemical vapor deposition (HVCVD) using benzylamine as both carbon and nitrogen sources. Nitrogen atoms were successfully doped into double-walled carbon nanotubes (DWCNTs) using the HVCVD method in one step without further functionalization. Purified N-DWCNTs were coated with PVP. The results reveal that as-fabricated sensors from PVP-coated N-DWCNTs enabled a 15-fold improvement in formaldehyde detection compared with un-doped DWCNTs. The LOD of PVP-coated N-DWCNTs was 15 ppm. Although the LOD of PVP-coated N-DWCNTs sensor was much higher compared with previous reports [12,13,14,15], PVP-coated N-DWCNTs sensor can be used down to ppm-level formaldehyde detection at room temperature. The experimental study of pristine CNTs and their functionalization or composite materials for formaldehyde detection is summarized in Appendix A. The fabricated hybrid sensor based on PVP-coated N-DWCNTs shows advantages over previous reports in terms of the enhancement of the sensitivity and selectivity to formaldehyde by synergistic effect of via polymer swelling and nitrogen doping in DWCNTs. 

## 2. Materials and Methods

DWCNTs and N-DWCNTs were synthesized by HVCVD at 950 °C using ethanol and benzylamine, respectively, as carbon and nitrogen sources. As-synthesized DWCNTs and N-DWCNTs were purified by acid and air-annealing treatments three times, similar to the previous report [20]. After the purification process, purified DWCNTs and N-DWCNTs were separately dispersed for 60 min in ethanol by ultrasonic dispersion. The concentration of DWCNTs and N-DWCNTs in ethanol was 0.1 mg/mL. 25 µL of DWCNTs-ethanol and N-DWCNTs-ethanol were dropped separately onto the printed circuit board substrate, which consists of an interdigitated Cu/Au electrode. During drop casting, the sensor substrate was heated to 100 °C to remove the solvent in the sensing material. Next, PVP (Mw = 40,000 g/mol) were diluted in toluene and stirred for 24 h. The PVP concentration was fixed at 3 wt.%. The PVP solution was then dropped onto pre-dropped DWCNT and N-DWCNTs on the sensing substrate, which was then heated to 100 °C for 24 h to remove the solvent in the sensing materials (hereafter referred to as PVP/DWCNTs and PVP/N-DWCNTs, respectively). Field emission scanning electron microscopy (FESEM, JEOL JSM-7800F) operating at 1 keV, transmission electron microscopy (TEM, JEOL JE-2010) operating at 200 kV and Raman spectroscopy (NT-MDT, NTEGRASPECTRA) using an Ar laser light source with a wavelength of 532 nm were used to analyze the morphology, nanostructure, and crystallinity of all the samples. The chemical composition and state of DWCNTs and N-DWCNTs was characterized by X-ray photoelectron spectroscopy (XPS) using hemispherical SCIENTA RS4000 photoelectron equipped with a monochromatic Al Kα radiation (1486.7 eV). 

Sensor response (the changing of electrical resistance) was measured before and after exposure to VOCs. The electrical resistance of the sensing material between the interdigitated electrodes was measured using a multimeter (UNI-T; UT71D). For comparison, the sensing performance, i.e., the percentage of the sensor response (*SR%*), can be calculated using Equation (1), whereas R_VOC_ and R_air_ are the electrical resistance of the sensor under VOC vapor and air atmosphere, respectively. As-fabricated sensors from DWCNTs, N-DWCNTs, PVP/DWCNTs and PVP/N-DWCNTs were placed separately in stainless steel chamber under air atmosphere for 3 min, followed by the dropping of VOCs liquid into the detection chamber, and the electrical resistance was observed for 3 min. Finally, all sensors were recovered in an air atmosphere for 5 min. All the measurements were carried out at room temperature (25 ± 2 °C) and relative humidity of 52 ± 2%. The concentration of VOCs was controlled by the VOC volume compared with the detection chamber. The concentration of VOCs varied in the range of 100–5000 ppm. The VOCs were 1,2-dichloroethane, dichloromethane, formaldehyde, and toluene.
SR (%) = ((R_VOC_ − R_air_)/R_air_) × 100%.(1)

## 3. Results

### 3.1. Characterization: Nanostructure, Morphology Crystallinity, and Chemical Composition

Nanostructure, morphology, crystallinity, and chemical composition were performed by TEM, FESEM, Raman spectroscopy and XPS, respectively. Figure 1 shows the TEM images of (a–c) DWCNTs and (d,e) N-DWCNTs in a bundle and individual structures. TEM observation reveals the tubular structures composed of double rolled-up graphene. The size of DWCNTs is in the range of 14 to 33 nm, as shown in Figure 1a. Figure 1b shows a cross-section, high-resolution TEM image of a bundle of DWCNTs with a close-packed arrangement of the DWCNTs. Figure 1c and e display the isolated DWCNTs and N-DWCNTs, respectively. The average diameters of DWCNTs and N-DWCNTs were 1.88 ± 0.61 and 1.87 ± 0.30 nm, respectively. The diameter of N-DWCNTs decreased slightly compared with that of undoped DWCNTs. Figure 1f reveals an entangled network of N-DWCNTs on the substrate. After the drop-casting of PVP on N-DWCNTs, the DWCNT network was partially covered with PVP, as shown in Figure 1g. Furthermore, an increase in the interspacing of N-DWCNT network was found, as shown in Figure 1g. 

Figure 2a shows the radial breathing mode (RBM) of the DWCNTs and N-DWCNTs and Figure 2b shows the Raman spectra of all samples. DWCNTs and N-DWCNTs are composed of three main peaks, namely radial breathing mode (RBM) at 145–275 cm^−1^, disorder-band (D-band) at ~1340 cm^−1^, and graphite-band (G-band) at ~1590 cm^−1^ [20]. The diameter of DWCNTs can be estimated using Equation (2), where ω_RBM_ and d_t_ are the RBM frequency (cm^−1^) and the tube diameter (nm), respectively [21,22]. The calculated diameters of DWCNTs from RBM peaks at 146, 188, 231 and 271 cm^−1^ were 1.72, 1.31, 1.06 and 0.90 nm, respectively, while that of N-DWCNTs at 142, 160, 175,193, 218, 262 and 287 cm^−1^ were 1.77, 1.56, 1.42, 1.28, 1.13, 0.93 and 0.84 nm, respectively. The interlayer spacing of DWCNTs and N-DWCNTs is in the range of 0.33–0.36 nm, which is consistent with the interlayer distance in graphite [23,24]. The calculated diameter of the DWCNTs is in the same range as that observed from the TEM results.
ω_RBM_ = 234/d_t_ + 10. (2)

The intensity ratio of D-band and G-band (*I_D_*/*I_G_*) of DWCNTs and N-DWCNTs were 0.096 and 0.124, respectively. The higher *I_D_*/*I_G_* of N-DWCNTs compared to that of DWCNTs implied that nitrogen atom can be incorporated in the DWCNT structure, resulting in the formation of defects, thus increasing the value of *I_D_*/*I_G_* ratio. After the drop-casting of PVP on the DWCNTs and N-DWCNTs, the *I_D_*/*I_G_* of PVP/DWCNTs and PVP/N-DWCNTs were 0.1083 and 0.1327, respectively. The *I_D_*/*I_G_* slightly increases, implying that coating DWCNTs and N-DWCNTs with PVP did not affect their intrinsic structure significantly. XPS was used to confirm the quantity of the nitrogen atom. In XPS survey analysis, DWCNTs and N-DWCNTs consist of two main peaks, namely C 1s and O 1s peaks at ~284 eV and ~533 eV, respectively, as shown in Figure 3a. The C 1s peak comprises three carbon statuses: sp^2^ C=C at 284.8 eV, C-O bands at 286.2 eV and shake-up satellite line at 290.5 eV [25,26]. Figure 3b reveals that the N-DWCNTs comprises four different types of nitrogen: pyridinic at 398.2 eV (0.36 at%), pyrrolic at 399.9 eV (0.53 at%), graphitic at 401.5 eV (0.23 at%) and oxidized at 403.3 eV (0.19 at%) [26]. N-DWCNTs contain approximately 1.31% of nitrogen in total. The C 1s, O 1s and N 1s composition of DWCNTs and N-DWCNTs is summarized in Table 1. Raman spectroscopy and XPS analysis confirm the existence of nitrogen doping of the N-DWCNTs. 

### 3.2. Gas Sensing Characteristics 

Figure 4a–d shows the sensor response of all the sensors as a function of time under exposure to 1,2-dichloroethane, dichloromethane, toluene, and formaldehyde vapors at 5000 ppm. It was found that the electrical resistance of all sensors increased under VOC vapor and decreased under air. The electrical resistance of some sensors was unable to recover to its initial electrical resistance due to the remaining VOC molecules in the sensing materials. To completely remove VOCs, we suggest the integration of a heating system with a sensor substrate, or an ultraviolet irradiation system which could eliminate the remained VOC. For the sensing mechanism, the VOC with a dipole moment makes the coulomb force between VOC and p-type CNTs, leading to the creation of a depletion layer due to the holes being interrupted in the p-type CNTs, resulting in an increased electrical resistance of the sensing materials [27]. 

Figure 5a shows the sensor response of all the sensors exposed to all VOCs at 5000 ppm. The results show that the as-fabricated PVP/DWCNTs and N-DWCNTs sensors show a 2.0-fold and 8.7-fold increase in the response to 5000 ppm formaldehyde vapor, respectively, in comparison to that of un-doped DWCNTs, and, conversely, a decrease in the sensor response to 1,2-dichloroethane, dichloromethane and toluene. Except for the PVP/DWCNTs, the sensor response increases slightly for dichloromethane detection. The results imply that sensor response to formaldehyde is improved by polymer functionalization and nitrogen doping. For the sensing mechanism of PVP/DWCNTs, a polymer swelling model is proposed. When the VOC vapor adsorbed on the surface of PVP, the VOC molecules diffused into the PVP. The PVP is swelled due to the dissolution of VOC molecules in the PVP. The polymer swelling broke and loosened the conductive path of CNTs network, which resulted in an increase in electrical resistance [28,29,30], as shown in Figure 6. In the case of N-DWCNTs, the greater sensing performance of N-DWCNTs to formaldehyde vapor is attributed to the presence of a nitrogen-site on the surface of DWCNTs, which increases the strong interaction between formaldehyde molecule and the nitrogen site [18,31]. Moreover, the electrical resistance of N-DWCNTs is unable to recover to its initial value under air exposure (as shown in Figure 4d). We envision that formaldehyde molecules strongly interact with nitrogen sites, resulting in difficulty in removing the formaldehyde molecules from the N-DWCNTs. A previous theoretical study reported that nitrogen-doped single-walled carbon nanotubes (N-SWCNTs) show high sensitivity to formaldehyde molecule compared to un-doped SWCNTs at room temperature. N-SWCNTs show a remarkable change in energy gap after the adsorption of formaldehyde. The energy gap of N-SWCNTs decreases after adsorption of the formaldehyde molecule. This means a stronger interaction between the N-SWCNTs, thus a greater response, compared with that of the un-doped SWCNTs [32]. Our experimental results are consistent with the theoretical calculations. On the other hand, the sensor responses of N-DWCNTs to the 1,2-dichloroethane, dichloromethane, and toluene vapors significantly decrease by approximately 3.4, 15.9 and 7.3 times, respectively, which is lower than that of the DWCNTs. It is demonstrated that 1,2-dichloroethane, dichloromethane and toluene adsorb unfavorably on the nitrogen-site in N-DWCNTs. These molecules are easily removed by under air exposure (as shown in Figure 4a–c). Remarkably, the as-fabricated PVP/N-DWCNTs sensor exhibits high sensitivity and selectivity toward formaldehyde vapor. The PVP/N-DWCNTs enabled 15-fold improvement in formaldehyde detection compared to the un-doped DWCNTs under room-temperature conditions. The greater sensing performance of PVP/N-DWCNTs to formaldehyde vapor is attributed to the synergistic effect of the polymer swelling and nitrogen-site in N-DWCNTs. 

Next, all sensors were further applied to the formaldehyde at 250–5000 ppm to investigate the sensor response pattern and LOD [33]. Figure 5b shows the sensor response of all the sensors as a function of formaldehyde concentration. It was found that the responses of DWCNTs and PVP/DWCNTs increased in a linear fashion from 0.20 to 2.28% and 0.23 to 4.53%, respectively, with formaldehyde concentration in the range of 1000–3000 ppm. We anticipate that the DWCNTs and PVP/DWCNTs sensors were unable to detect formaldehyde with concentration less than 1000 ppm. For N-DWCNTs and PVP/N-DWCNTs sensors, the sensor response of N-DWCNTs increases in a linear fashion from 7.83 to 14.57%, respectively, with formaldehyde concentration in the range of 250 to 1000 ppm, and tends to saturate the response from 14.57 to 19.82% for formaldehyde at 1000–3000 ppm, while that of the sensor response of PVP/N-DWCNTs increases rapidly from 9.50 to 27.70% in the concentration range of 250–3000 ppm and tends toward saturation from 27.70 to 33.72% at 3000–5000 ppm. The calculated LODs of DWCNTs, PVP/DWCNTs, N-DWCNTs and PVP/N-DWCNTs are 506, 410, 115 and 15 ppm, respectively. The LOD of the PVP/N-DWCNTs shows 34-fold higher than that of the DWCNTs. These results demonstrate that the fabricated sensor from N-DWCNTs can be used down to ppm-level formaldehyde detection at room temperature. Future work will study the decoration of PVP/N-DWCNTs with various metal nanoparticle for improving the sensitivity and selectivity towards formaldehyde down to ppb level. To investigate the reproducibility, PVP/N-DWCNTs sensors were repeatedly exposed to formaldehyde at 250 ppm for four cycles, as shown in Figure 5c. The response of the sensor is sustained, implying the good repeatability of PVP/N-DWCNTs sensors.

## 4. Conclusions

In this study, we demonstrated a high-performance formaldehyde sensor based on PVP/N-DWCNTs hybrid materials. The as-fabricated PVP/N-DWCNTs sensor exhibits 15-fold improvement in the sensor response to formaldehyde detection compared to the DWCNTs sensor. The enhancement of the sensitivity and selectivity is achieved by PVP swelling and strong interaction between the nitrogen sites on the surface of N-DWCNTs and the formaldehyde. The calculated detection limit of as-fabricated PVP/N-DWCNTs sensor was estimated to go down to 15 ppm, which is 34 times higher than that of the as-fabricated DWCNTs sensor. This study demonstrates the possibility of PVP/N-DWCNTs hybrid materials as sensing materials for practical and high-performance formaldehyde sensing at room temperature. 

## Figures and Tables

**Figure 1 sensors-22-09329-f001:**
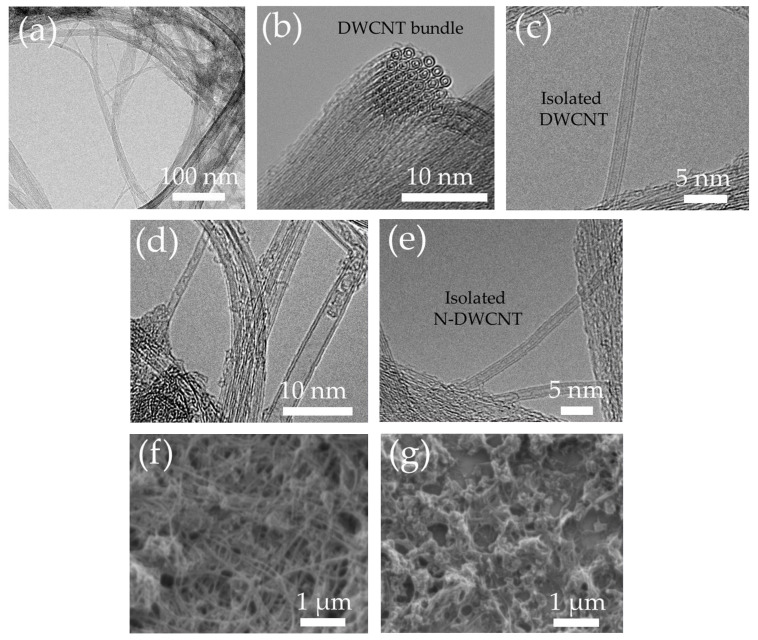
TEM images of (**a**–**c**) DWCNTs and (**d**,**e**) N-DWCNTs. FESEM images of (**f**) N-DWCNTs and (**g**) PVP/N-DWCNTs on substrate.

**Figure 2 sensors-22-09329-f002:**
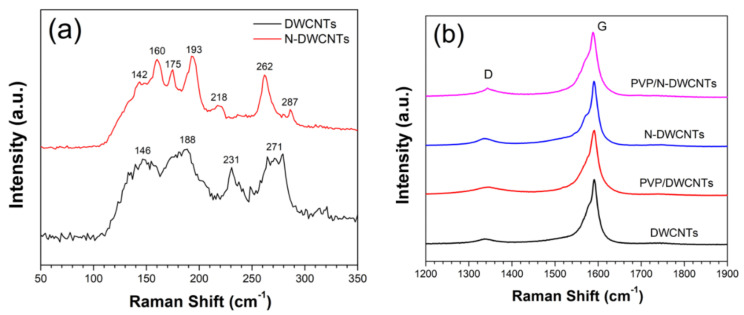
(**a**) RBM peaks of the DWCNTs and N-DWCNTs. (**b**) Raman spectra of all the samples.

**Figure 3 sensors-22-09329-f003:**
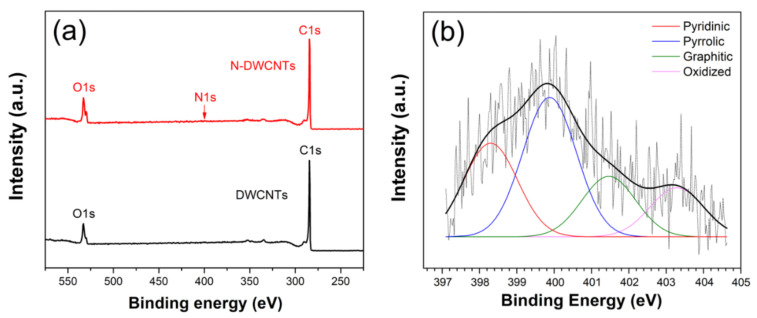
(**a**) XPS survey spectra of the DWCNTs and N-DWCNTs. (**b**) N 1s spectrum and peak fitting of the N-DWCNTs.

**Figure 4 sensors-22-09329-f004:**
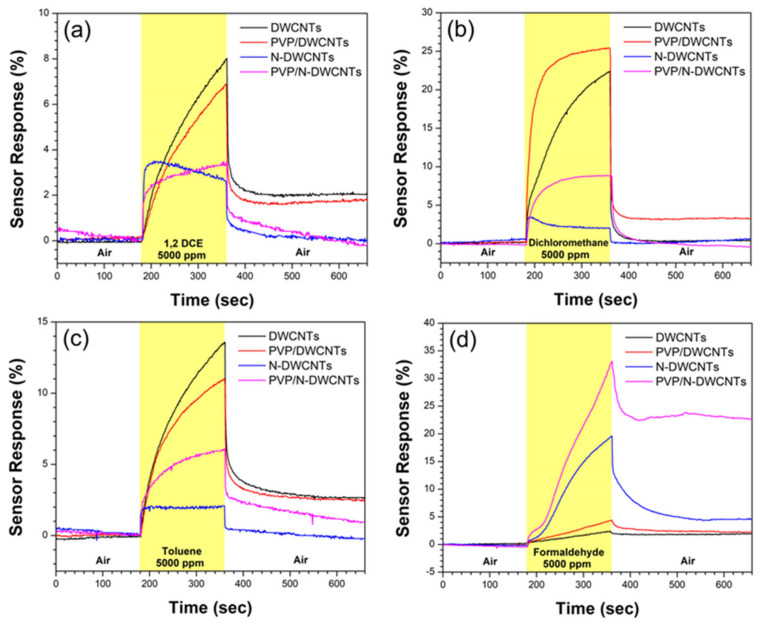
Sensor response of all sensors as a function of time exposed to (**a**) 1,2-dichloroethane, (**b**) dichloromethane, (**c**) toluene, and (**d**) formaldehyde at 5000 ppm.

**Figure 5 sensors-22-09329-f005:**
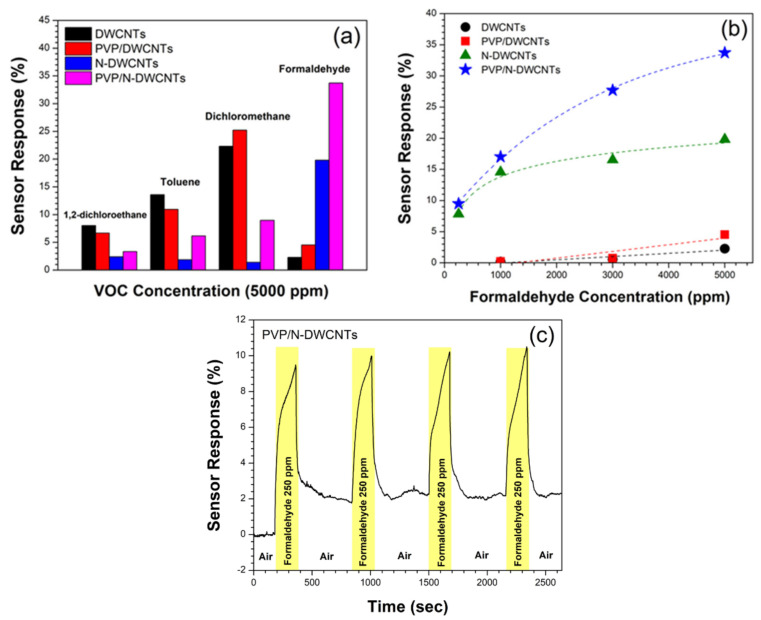
(**a**) Sensor response of all the sensors exposed to different VOCs at 5000 ppm. (**b**) Sensor responses of all the sensors as a function of formaldehyde concentration. (**c**) Reproducibility of the response of PVP/N-DWCNTs sensors to 250 ppm formaldehyde.

**Figure 6 sensors-22-09329-f006:**
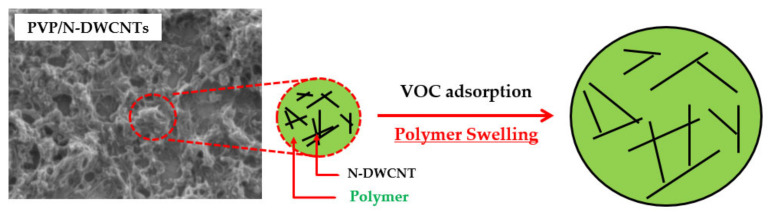
Schematic diagram of gas sensing mechanism of PVP/N-DWCNTs for VOC detection.

**Table 1 sensors-22-09329-t001:** The C 1s, O 1s and N 1s composition of DWCNTs and N-DWCNTs.

Sample	Content (at%)			N/C Ratio (at%)
	C	O	N	
DWCNTs	88.11	11.89	-	-
N-DWCNTs	84.50	14.39	1.11	1.31

## Data Availability

Not applicable.

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
