# Peer review of "Highly Sensitive and Selective Formaldehyde Gas Sensors Based on Polyvinylpyrrolidone/Nitrogen-Doped Double-Walled Carbon Nanotubes"

_sensors, 2022, doi:10.3390/s22239329_

Round 1
Reviewer 1 Report
The manuscript of Chobsilp and coworkers is based on the fabrication of a hybrid material based on double-walled carbon nanotubes and polyvinylpyrrolidone for their use in sensing. Overall, the authors developed a sensor with high sensitivity and selectivity to formaldehyde but a low response to 1,2-dichloroethane, dichloromethane and toluene.
The reviewer has some suggestions and questions on this manuscript that would help increase the impact of this manuscript.
1. Up to now, more and more carbon nanotubes sensors have been prepared to detect volatile organic compounds, including formaldehyde. What is new compared with already published papers? The merits and significance of this work should be stressed.
2. The elemental analysis data should be provided in a Table, together with the exact composition of the investigated sensors.
3. The authors claim that sensor response to formaldehyde was improved by polymer functionalization. But why is this effect not observed in all the studied VOCs or in the DWCNTs materials?
4. A critical comparation of the limit of detection with the already published papers detecting formaldehyde should be placed.
5. What about the reproducibility of the sensor response (i.e., recyclability)?
Author Response
Please find all my answers in the attached file (Response to Reviewers 1)

Reviewer 2 Report
In this manuscript (sensors-2012300), the formaldehyde sensors are fabricated using hybrid materials of nitrogen-doped double-walled carbon nanotubes (N-DWCNTs) and polyvi-24 nylpyrrolidone (PVP). Gas sensing results are good and progressive, and the topic can attract a wide range of readerships. But there are some problems in the introduction, presentation, and discussion of the results. As such, some revisions are needed before possible publication. My specific comments are as follows:
1. Introduction: The motivation of PVP as a decorative material needs to be discussed.
2. Equation (1): The percent sign should be after 100 (100%). The definition of response needs literature support, referring to Sens. Actuators B Chem. 363 (2022) 131790.
3. Gas sensing test: Detailed test conditions should be given, such as method, equipment and relative humidity.
4. Results (Figure 4): In order to analyze the mechanism conveniently, it is suggested to provide the corresponding actual resistance response curve.
5. As a research paper, it is recommended to provide a detailed gas sensing mechanism (combine the mechanism diagram).
6. The figures’ quality needs to be improved, such as line width and font size.
7. Check writing, for example, detection (LOD) [31] Fig. 5(b) shows.
Author Response
Please find all my answers in the attached file (Response to Reviewers 2)

Round 2
Reviewer 1 Report
The authors did not consider adding to the revised version a critical comparison of the limit of detection with the already published papers detecting formaldehyde, as the reviewer asked. Why?
As the author comment, the reproducibility of a sensor response is one of the critical factors and must be studied to show that the manuscript has some reliable information. The authors failed to prove that they repeated the measurements.
